**DOI: 10.1038/s41467-018-05784-3**　　**OPEN**

# Single-cell analysis reveals that stochasticity and paracrine signaling control interferon-alpha production by plasmacytoid dendritic cells

Florian Wimmers [1,2], Nikita Subedi[3,4], Nicole van Buuringen[1], Daan Heister[1], Judith Vivié[5],
Inge Beeren-Reinieren[1], Rob Woestenenk[6], Harry Dolstra[6], Aigars Piruska[7], Joannes F.M. Jacobs[8],
Alexander van Oudenaarden[5], Carl G. Figdor[1], Wilhelm T.S. Huck[7], I. Jolanda M. de Vries[1] & Jurjen Tel [1,3,4]

Type I interferon (IFN) is a key driver of immunity to infections and cancer. Plasmacytoid dendritic cells (pDCs) are uniquely equipped to produce large quantities of type I IFN but the mechanisms that control this process are poorly understood. Here we report on a droplet-based microfluidic platform to investigate type I IFN production in human pDCs at the single-cell level. We show that type I IFN but not TNFα production is limited to a small sub-population of individually stimulated pDCs and controlled by stochastic gene regulation. Combining single-cell cytokine analysis with single-cell RNA-seq profiling reveals no evidence for a pre-existing subset of type I IFN-producing pDCs. By modulating the droplet micro-environment, we demonstrate that vigorous pDC population responses are driven by a type I IFN amplification loop. Our study highlights the significance of stochastic gene regulation and suggests strategies to dissect the characteristics of immune responses at the single-cell level.

[1] Department of Tumor Immunology, Institute for Molecular Life Sciences, Radboud University Medical Center, Nijmegen 6525 GA, The Netherlands. [2] Institute for Immunity, Transplantation and Infection, Stanford University, Stanford 94305 CA, USA. [3] Department of Biomedical Engineering, Laboratory of Immunoengineering, Eindhoven University of Technology, Eindhoven 5612 AZ, The Netherlands. [4] Institute for Complex Molecular Systems, Eindhoven University of Technology, Eindhoven 5612 AZ, The Netherlands. [5] Hubrecht Institute - KNAW and University Medical Center Utrecht, Utrecht 3584 CT, The Netherlands. [6] Department of Laboratory Medicine, Laboratory of Hematology, Radboud University Medical Center, Nijmegen 6525 GA, The Netherlands. [7] Department of Physical Organic Chemistry, Institute for Molecules and Materials, Radboud University, Nijmegen 6525 HP, The Netherlands. [8] Department of Laboratory Medicine, Laboratory Medical Immunology, Radboud University Medical Center, Nijmegen 6525 GA, The Netherlands. These authors contributed equally: Carl G. Figdor, Wilhelm T.S. Huck, I. Jolanda M. de Vries.  Correspondence and requests for materials should be addressed to J.T. (email: J.Tel@tue.nl)

Plasmacytoid dendritic cells (pDCs) are blood circulating innate immune cells with the unique ability to rapidly release large quantities of type I interferon (IFN) for antiviral immunity[1–3]. pDC-produced type I IFN is associated with effective anti-cancer immunity but is also a driver of autoimmune diseases[4–8]. Type I IFN production by pDCs is initiated when nucleic acids trigger the endosomal Toll-like receptors (TLRs) 7 or 9 leading to the activation of transcription factor interferon regulatory factor-7 (IRF7), which only pDCs express constitutively and at high levels[9–11]. Several pDC subclasses were proposed and single-cell genomic profiling revealed ample variation in the molecular outfit of individual DCs[12–16]. These individual differences may have an impact on the ability of each pDC to produce type I IFN, and in non-pDC model systems random differences between virus-infected cell populations, attributed to stochastic gene regulation, caused significant variation in the production of type I IFN[17–21]. Additionally, type I IFN production by pDCs can be modulated by the microenvironment via soluble factors or cell surface receptors[22–27]. It is currently not known how pDC populations combine the complex information from TLR signaling and microenvironmental factors with random variations in the molecular outfit of individual pDCs to generate robust type I IFN responses. The question remains whether pDCs display stochastic expression of type I IFN despite high IRF7 expression, and whether pDC populations exploit environmental cues to counterbalance potential heterogeneity arising from this phenomenon.

Here, we developed a droplet-based microfluidic platform to dissect the human pDC-driven type I IFN response at the single-cell level within a tunable microenvironment. Generating thousands of identical droplets at high throughput allows massively parallelized single-cell experiments within these bioreactors. Recent technological breakthroughs in the field of droplet-based microfluidics increased the throughput of single-cell DNA and RNA-sequencing experiments by orders of magnitude[28,29]. Previous attempts by our lab and others to leverage this power for the analysis of cytokine secretion were hampered in their translation into practice due to complex detection equipment or difficult handling conditions[30,31]. Here, we demonstrate the detection of cytokine secretion and activation marker expression by individually stimulated cells in droplets and reveal stochastic differences in pDC-driven type I IFN production. Single-cell RNA-sequencing (ScRNA-seq) of these cells allowed us to profile the transcriptional changes in each cell upon perturbation with TLR ligands and links transcriptional variation to cytokine secretion at the protein level. Finally, by varying key droplet parameters, we find that single pDCs collaborate to amplify their activity and generate population-driven type I IFN responses.

## Results

### Functional pDC heterogeneity arises early after stimulation.

pDCs operate in complex microenvironments that influence their cellular state. To investigate the intrinsic potential of single pDCs to produce IFNα without interference of other cells, we developed a droplet microfluidic single-cell assay for the detection of cytokine secretion (Fig. 1a). In short, pDCs were coated with capture reagents for cytokine readout and encapsulated in picoliter droplet microenvironments using a microfluidic device (Fig. 1b, c). During in-droplet incubation, produced IFNα and tumor necrosis factor-α (TNFα) was captured on the cell surface by the cytokine capture reagents. After breaking the emulsion, pDCs were isolated and analyzed via multicolor flow cytometry. Each droplet served as a standardized and independent cell reactor and allowed the investigation of tens of thousands of individually stimulated cells simultaneously. This massively parallel approach facilitated

the characterization of rare, truly single-cell behavior. This system greatly exceeds the throughput and possibilities when compared to conventional limited dilution experiments which require numerous replicate cultures and, crucially, cannot prohibit cellular crosstalk. Further, the low droplet volume greatly reduced reagent consumption and allowed us to work with small numbers of (primary) cells. We routinely probed rare pDCs using as few as 40,000 cells as input, showing that our technique is highly suited for the use of small biological samples. Importantly, our droplet-based cytokine capture approach enables sensitive cytokine detection and makes no use of transport inhibitors, which negatively impact cellular function and viability. This enabled us to measure cytokine secretion for extended time periods in an accumulative rather than snapshot fashion and facilitated the analysis of extremely early secretion events within the first 30 min of stimulation. Early activation events are problematic to investigate with transport inhibitor-based methods as they negatively impact cell signaling, thereby distorting the measurement. In contrast to microtiter-based approaches, our microfluidic setup makes use of computer-controlled syringe pumps. This allowed us to precisely control environmental factors and vary droplet volume and local cell density in a range that currently cannot be obtained with conventional cultures.

First, we encapsulated pDCs in picoliter droplets (average 92 pL, SEM 1.8 pL, $n = 85$) with an encapsulation efficiency of approximately 6% cell-containing drops (Fig. 1d) of which 96% contained a single cell (Fig. 1e). Cells were incubated with the synthetic nucleic acid compound CpG-C (TLR9 agonist) and analyzed by flow cytometry (Fig. 1f). Strikingly, only a minor subset of pDCs produced IFNα, which emerged as early as 2 h after stimulation and peaked after 6 h of stimulation (Fig. 1g, h). In contrast, we observed that virtually all pDCs produced TNFα during incubation in droplets (Fig. 1g, h). Similarly, the majority of pDCs was positive for the activation markers CCR7, CD40, and CD86 and most pDCs were highly multifunctional (Fig. 1i–k). Furthermore, we confirmed previous findings that a recently discovered subset of pDC-like progenitor cells, called AS DCs, was not involved in IFNα production (Supplementary Fig. 1)[15].

Next, we studied the capacity of TLR signaling to modulate the probability of pDCs to produce IFNα. We encapsulated cells with varying concentrations of CpG-C and measured the fraction of cells producing IFNα (Fig. 2a). Surprisingly, we only observed minor variations in the fraction of pDCs secreting IFNα irrespective of the concentration of CpG-C. In contrast, the production of TNFα, the expression of the activation markers CCR7, CD40, and CD86, and cell viability all positively correlated with CpG-C concentration (Fig. 2a, Supplementary Fig. 2). To exclude CpG-C-specific limitation in the TLR9 signaling pathway, we stimulated pDCs with the synthetic TLR7/8-agonist R848 (Fig. 2b) and the strong IFNα inducer TLR9 agonist CpG-A (Supplementary Fig. 3). Similar to CpG-C stimulation, only a small fraction of pDCs produced IFNα and this effect was independent of stimulus concentration. Thus far, virtually all knowledge on IFNα secretion by human pDCs is based on bulk cultures. Therefore, cells from the same donor were analyzed by microfluidics and bulk culture side-by-side. Our results demonstrate that individually stimulated cells in droplets indeed have an inferior capacity to secrete IFNα as compared to bulk stimulated cells (Supplementary Fig. 4). Finally, to rule out that the observed IFNα production was due to stimulus-independent constitutive secretion, we stimulated pDCs with either interleukin-3 (IL-3) or CpG-C. The pDCs treated with IL-3, a survival factor for pDCs, which only survive briefly ex vivo when left unstimulated, showed a significantly reduced probability to produce IFNα (Fig. 2c). To exclude that IFNα production by single pDCs is delayed compared to bulk analysis, we incubated pDCs for 12 h and 24 h but only observed small deviations (Fig. 2d).

Together, our data demonstrate that our microfluidic assay is suited for the sensitive detection of cytokine secretion and protein expression by single cells. Functional heterogeneity emerges immediately after TLR activation in pDCs, as only a small fraction of cells is able to produce IFNα. IFNα production is enhanced by TLR signaling but appears to be regulated by an additional stochastic, i.e. random, component which is not associated with strength, amplitude, or duration of cell activation.

**Type I IFN is an important regulator of early pDC function.** Cellular heterogeneity often emerges from random processes during gene transcription[32]. To probe whether the observed differences in IFNα production originate from such stochastic gene regulation or whether a privileged pDC subset already exists at steady state, we employed scRNA-seq to profile the onset of the type I IFN response upon perturbation with CpG-C. Freshly isolated pDCs from a healthy donor were encapsulated in droplets and individually stimulated with CpG-C using our

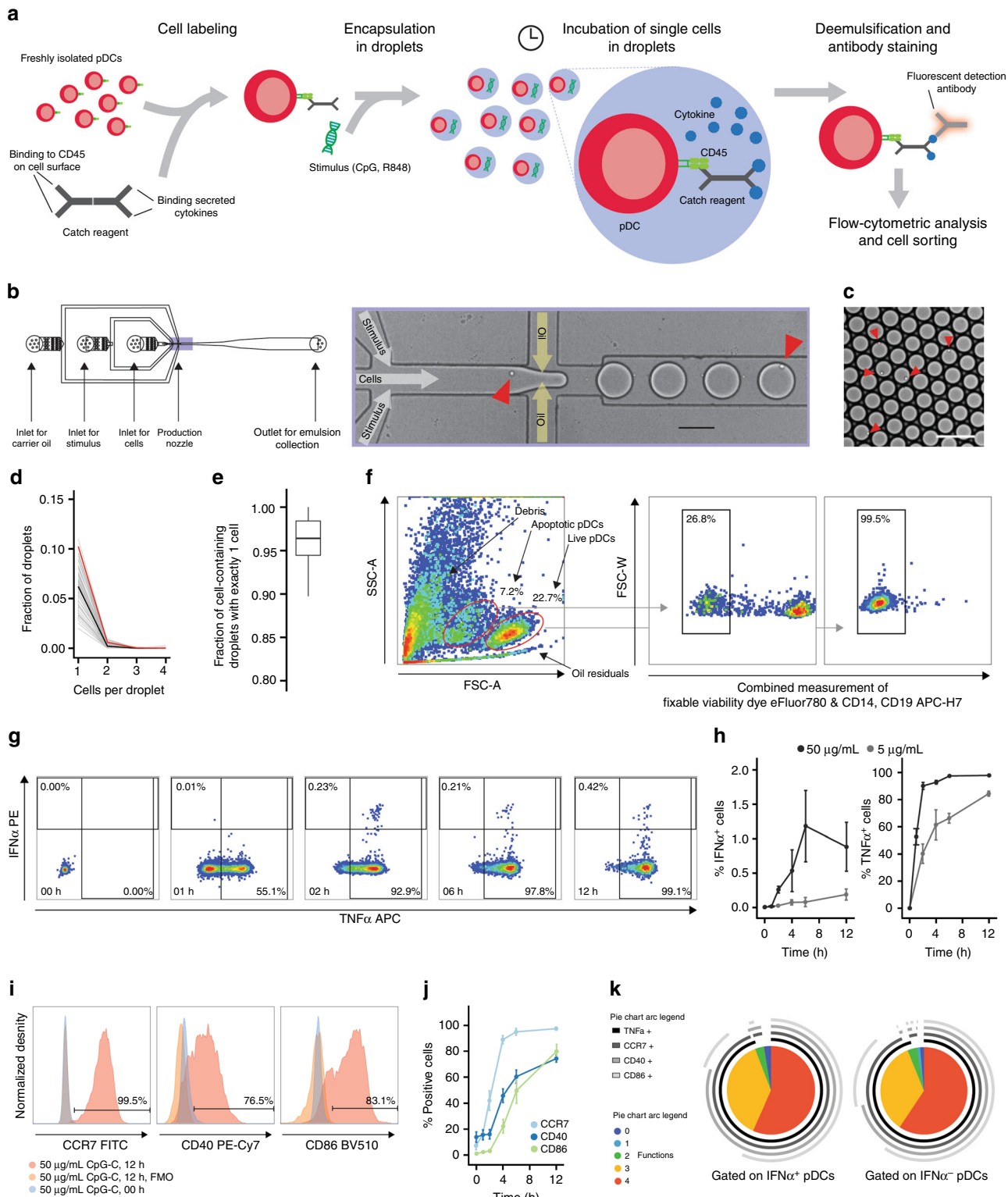

microfluidic platform (Fig. 3a). After incubation for 0, 1, or 2 h, the emulsion was broken, cells were stained for cytokine secretion, and 384 cells at each time point were sorted into well plates for scRNA-seq (Fig. 3b). Single cells were processed using the SORT-Seq (sorting and robot-assisted transcriptome sequencing) protocol followed by sequencing of ~0.1 million to 0.2 million paired-end reads per cell[33]. At 2 h, when we detected the first IFNα$^+$ pDCs, we enriched for this subset by sorting 39 IFNα$^+$ pDCs before randomly filling up remaining wells with IFNα$^+$ and IFNα$^-$ cells. In total, we profiled 1152 cells with an average of 4677 transcripts per cell and 1574 unique genes detected per cell. After filtering, down-sampling, and removal of 141 DCs that clustered separately in initial analyses and expressed gene signatures of non-pDC subsets (CD1c$^+$, CD141$^+$), the final dataset contained 774 cells expressing 13,214 genes (Supplementary Fig. 5)[15].

Unsupervised *k*-medoid clustering of the correlation matrix combined with outlier detection using the raceID2 algorithm suggested the presence of 8 cell clusters (Supplementary Figure 6A–E) which were visualized in two dimensions using t-distributed stochastic neighbor embedding (t-SNE) (Fig. 3c, d; Supplementary Fig. 6F–H)[34]. We observed two clusters of unstimulated cells, Cl1 and Cl7. Cl1 also contained a group of cells that expressed characteristics of the described CD2$^{hi}$ pDC and AS DC subsets (Supplementary Fig. 7)[12,15]. Cells stimulated for 1 h mapped into Cl2 with few cells also present in Cl3 and Cl8. Cl4, Cl6, and Cl5 were dominated by pDCs that were stimulated for 2 h. Cells sorted as IFNα$^+$ mapped to Cl4 and Cl5 at equal fractions and more than 60% of cells in Cl5 were sorted as IFNα$^+$ (Fig. 3e). The pDCs mapping to Cl5 produced high levels of IFNα as measured by flow cytometry (Fig. 3f) and cells sorted as IFNα$^+$ expressed high levels of type I IFN genes, such as IFNA2, and IFNB1, as well as the interferon-inducible gene IFIT2 (Fig. 3g). Differential gene expression analysis showed an enrichment of type I IFN genes or type I IFN-induced genes in Cl5 as compared to all other cells (Fig. 3h). In contrast, no obvious transcriptionally distinct pDC subset that could predict type I IFN production was observed at steady state. This could either be because type I IFN production is genuinely a stochastic process, or because the nature of such a privileged cell state cannot be determined a priori by present technology. Similar results were obtained when pDCs from two additional healthy donors were profiled at steady state (Supplementary Fig. 8).

Next, we compared the gene expression of individually stimulated pDCs and unstimulated cells. We argued that the underlying mechanisms of Cl5 pDCs' unique activation state might become evident when comparing the differential gene expression profiles of all stimulated pDC clusters. On average, Cl5

pDCs showed 77 upregulated genes (log2(fold change) > 1.5; *p* value < 10$^{-8}$) and 1 downregulated gene (log2(fold change) < −1.5; *p* value < 10$^{-8}$) compared to unstimulated pDCs in Cl1 (Fig. 4a). Type I IFN and IFN response genes were among the most upregulated genes as well as several genes that support IFNα production in pDCs including MIR155HG, HSPA1A, and HSP90AA1 (Fig. 4b)[35−37]. Notably, the chemokines CCL3 and CCL4 that bind to the chemokine receptor CCR5 were upregulated. CCR5 is expressed on all pDCs and CCL3/4-CCR5 signaling might be responsible for the generation of large pDC clusters early after activation[38,39]. Next, we checked for other clusters with similar expression patterns. Cl5 cells shared many upregulated genes with cells from other clusters, especially Cl3 and Cl4; however, they also retained a group of uniquely upregulated genes centered around type I IFN production (Fig. 4c). Gene enrichment and functional annotation analysis using DAVID (Database for Annotation, Visualization and Integrated Discovery) showed that upregulated genes in Cl5 pDCs were enriched for anti-viral responses, cytokine responses, and apoptosis (Fig. 4d and Supplementary data 1−4)[40,41]. Similarly, type I IFN-related or -induced pathways were uniquely enriched in Cl5 genes, including TLR signaling, cytosolic DNA sensing, and the RIG-I-like pathway.

These results demonstrate that our microfluidic platform is ideally suited to work in conjunction with scRNA-seq to link functional information from extremely rare cells (<0.02% IFNα-producing pDCs) to whole transcriptome profiling. Together, the data show that type I IFN-producing cells possess unique transcriptional features, many of which are associated with autocrine type I IFN signaling. ScRNA-seq data revealed no evidence for a privileged pDC subset at steady state but type I IFN appears to be an important orchestrator of early pDC activation. The question remains of how pDC populations regulate the cellular heterogeneity originating from variation in type I IFN production.

**Environmental factors modulate type I IFN production.** In vivo, pDCs act in a dynamic microenvironment and migrate considerably during their life cycle. To assess the impact of environmental changes on the observed heterogeneity during pDC-driven type I IFN responses, we systematically varied key droplet parameters (Fig. 5a).

First, we generated droplets of varying size, covering several orders of magnitude (Fig. 5b). Single pDCs were encapsulated in droplets ranging from 31 to 1209 pL and stimulated for 12 h (Fig. 5c). No significant difference in the fraction of IFNα-secreting cells was detected (Fig. 5d, colored dots). Comparison with pDCs from additional donors, which were encapsulated in droplets of up to 3371 pL—a volume comparable to the average

**Fig. 1** Single-cell analysis reveals functional heterogeneity within individually stimulated pDCs. **a** Schematic overview of the droplet microfluidic assay. The pDCs were coated with cytokine capture reagents, encapsulated in picoliter droplets, and stimulated with TLR ligands. After incubation, cells were stained for viability, cytokine, and surface marker expression, and analyzed by flow cytometry. **b** Schematic overview of the employed microfluidic chip with microscopic image of the flow-focusing nozzle for the encapsulation of cells in droplets. **c** Microscopic image of emulsion after droplet production. **b**, **c** Red arrows indicate cells. Scale bars equal 100 μm. **d** The pDCs were encapsulated at a concentration of 1,300,000 cells/mL in 92 pL droplets. The distribution of cells in droplets was measured by manual analysis of microscopic images showing the emulsion directly after production. Shown is the fraction of droplets plotted against the number of cells per droplet; n = 85, black line indicates median, red line indicates predicted values. **e** Shown is the fraction of cell-containing droplets with exactly one cell; n = 85. Lines indicate mean, hinges mark interquartile ranges, and whiskers reach to the highest/lowest value that is within 1.5 × interquartile range. **f**–**k** The pDCs were treated as described above and stimulated with 5 μg/mL or 50 μg/mL CpG-C. **f** Viable pDCs were detected by forward scatter (FSC) and side scatter (SSC) analysis and subsequent gating on CD14$^-$CD19$^-$ and viability dye$^-$ cells. **g** IFNα- and TNFα-secreting cells were detected within that population. **h** Shown is the fraction of cytokine-secreting cells plotted against incubation time; n (5 μg/mL) = 3, n (50 μg/mL) = 6. **i** Surface marker-expressing pDCs were identified comparing the expression levels to fluorescence-minus-one controls. **j** Shown is the fraction of surface marker-expressing cells plotted against the incubation time; n > = 4. **k** The co-expression of CCR7, CD40, CD86, and TNFα by single IFNα$^+$ and IFNα$^-$ pDCs was analyzed. Shown is the relative contribution of each functional response pattern to the total pDC population. **h**, **j** Dots indicate mean, error bars indicate SEM

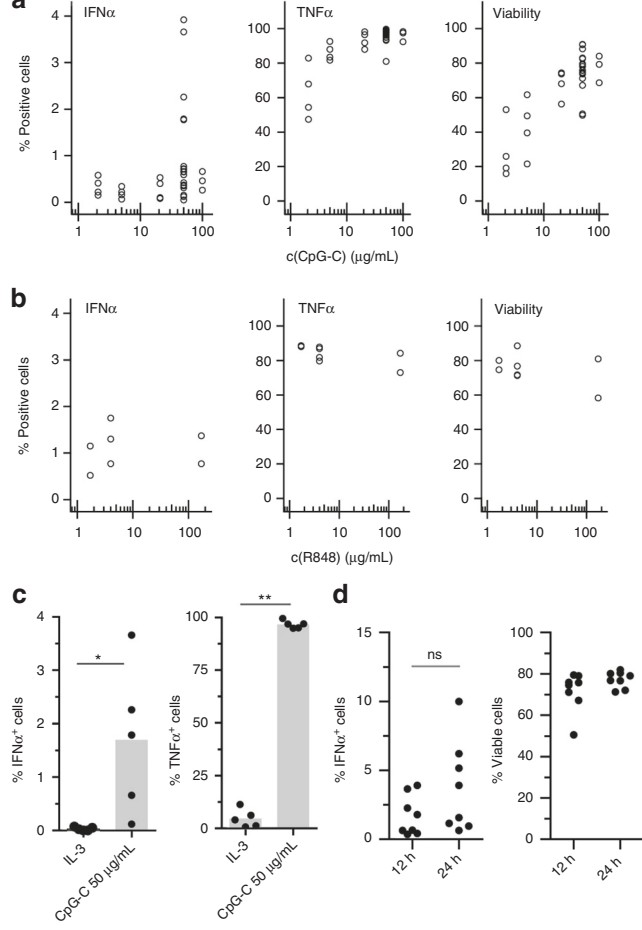

**Fig. 2** TLR-L concentration does not influence the fraction of IFNα-producing pDCs in droplets. **a**, **b** The pDCs were coated with capture reagent, encapsulated in picoliter droplets, and stimulated individually with **a** CpG-C or **b** R848 for 12 h. After staining for viability, surface marker expression and cytokine secretion, cytokine-secreting cells, and viable cells were detected via flow cytometry. Shown is the fraction of marker-expressing cells plotted against TLR ligand concentration. Different concentrations were tested in different donors; **a** $n \geq 3$, **b** $n \geq 2$. **c** The pDCs were treated as described above and stimulated with 0.01 µg/mL IL- 3 or 50 µg/mL CpG-C. Shown is the fraction of cytokine-secreting cells plotted against treatment condition; $n = 5$. Bars indicate mean. **d** The pDCs were treated as described above and stimulated with 50 µg/mL CpG-C for 12 h or 24 h. Shown is the fraction of IFNα-secreting or viable cells plotted against treatment condition; $n = 8$ (**c**, **d**) Mann–Whitney test *$p < 0.05$, **$p < 0.01$

volume of a single pDC in a perfectly mixed microtiter plate—showed no increase in the fraction of IFNα-secreting cells (Fig. 5d, gray dots).

Previous studies indicated that pDCs build homologous cell clusters upon stimulation, indicating that cellular crosstalk might be involved in their activation process[2,39]. To test this, we stimulated pDCs at various cell densities in microtiter plates in bulk (Supplementary Fig. 9A). Indeed, the fraction of IFNα-expressing pDCs depended on cell density (Supplementary Fig. 9B–F).

Communication between abovementioned pDCs in bulk can, thus, amplify the fraction of IFNα-producing cells. To get more insight into the nature of this communication, we tested whether crosstalk between two random interacting cells would be sufficient to enable IFNα production. We encapsulated pDCs in

90 pL droplets and gradually increased the fraction of droplets containing multiple cells (Fig. 5e). The fraction of IFNα-secreting cells increased slightly with the fraction of multiple cells per drop but did not match the predictions of a random interaction model (Fig. 5f, red). In contrast, the increase was better described by an alternative model based on the assumption that the early IFNα-producing pDCs activate co-encapsulated cells to produce IFNα as well (blue). However, in the employed system, we cannot rule out the possibility that the increase was caused by passive diffusion of cytokines or capture reagent between two co-encapsulated cells (Supplementary Fig. 10).

Together, these results show that the microenvironment—in this case represented by surrounding pDCs—has a critical impact on the probability of a pDC to produce IFNα.

**Priming with type I IFN increases the chance to produce IFNα.** Communication between IFNα-producing pDCs and surrounding pDCs occurs either in a juxtacrine or paracrine fashion. To elucidate whether paracrine signaling is the driving factor, freshly isolated pDCs were primed for 2 h with conditioned medium (CM) harvested from overnight bulk pDC cultures (Fig. 6a). After priming, pDCs were thoroughly washed, coated with capture reagent, encapsulated in picoliter droplets, and stimulated individually with CpG-C. Priming cells with 10% CM significantly increased the fraction of single pDCs that produced IFNα (Fig. 6b). In contrast, no effect was observed when cells were primed with remnant CpG-C, fresh cell culture medium, or primed without subsequent TLR stimulation. Similar to previous experiments, IFNα-secreting cells were highly multifunctional (Fig. 6c). Interestingly, the fraction of IFNα-secreting pDCs depended on the concentration of the CM, but not on the concentration of CpG-C (Fig. 6d, e). These findings indicate that priming modulates the probability of a pDC to produce IFNα but does not directly induce IFNα production.

To identify the responsible factor for the priming effect, we tested several cytokines described to positively affect IFNα-production by pDCs[22,27]. An initial screen revealed that only IFNβ—which acts similar to IFNα via the IFNα/β-receptor—increased the average per-cell IFNα production by pDCs cultured in microtiter plates at low cell density (Supplementary Fig. 11A, B). Furthermore, blocking the IFNα/β receptor and adding neutralizing antibodies against IFNα and -β inhibited the positive effect of CM on per-cell IFNα production. Priming of pDCs with IFNβ also led to an increase in the fraction of low-density cultured pDCs producing IFNα as measured by flow cytometry (Supplementary Fig. 11C). Finally, priming with IFNβ increased the fraction of IFNα-secreting cells in droplets to a similar extent as CM (Fig. 6f).

Previous studies identified IRF7 as a limiting factor in non-pDC models of type I IFN production[18]. In our hands, all pDCs displayed high levels of IRF7 immediately after isolation which decreased, however, during ex vivo culture (Supplementary Figs. 12, 13). High levels of IRF7 were induced by priming with IFNβ or by natural production of type I IFN by pDCs and coincided with but did not precede IFNα production (Supplementary Figs. 12, 13).

Together, these results unambiguously show that signaling via the type I IFN receptor amplifies TLR-induced IFNα production, thus modulating the patterns of heterogeneity within the pDC population. Next, we challenged our hypothesis that the bulk type I IFN response is governed by a small driver population of cells by conducting bulk experiments where type I IFN-mediated paracrine communication was abrogated by blocking the IFNα/β receptor and by adding neutralizing antibodies against IFNα and -β prior to bulk activation. Our experiments revealed that blocking the IFNAR in combination

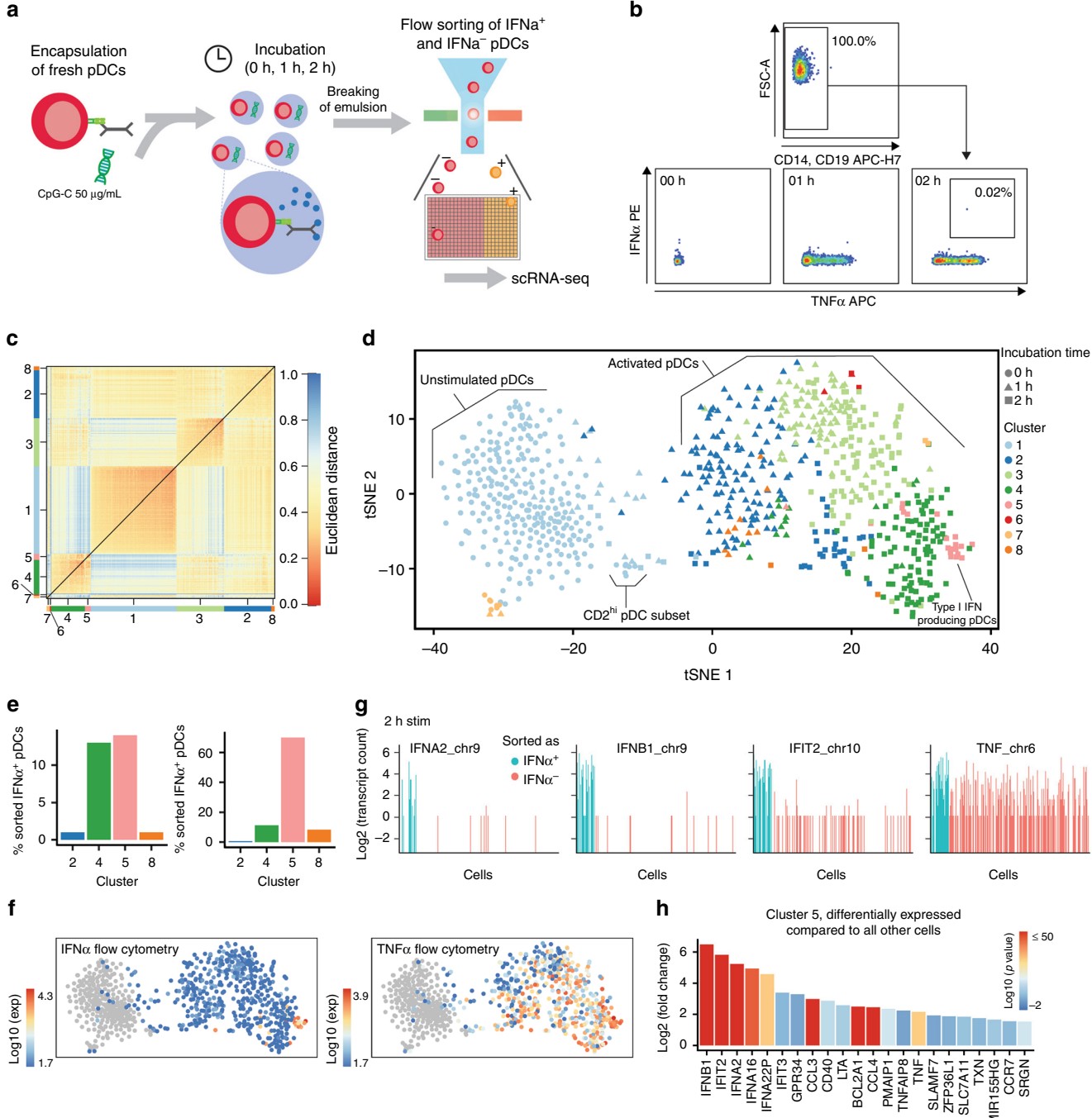

**Fig. 3** Single-cell RNA-sequencing identifies type I IFN-expressing cells early after activation. **a** The pDCs were coated with capture reagent, encapsulated in picoliter droplets, and stimulated individually with 50 μg/mL CpG-C for varying amounts of time. **b** After staining for surface marker expression and cytokine secretion, different pDC subsets were collected using fluorescence-activated cell sorting (FACS) and their transcriptomes were sequenced using the SORT-Seq protocol. **c** Heat map of the 774 cells that passed quality control filters representing transcriptome similarities as measured by Euclidean distance. The $k$-medoid clustering in combination with the raceID2 algorithm identified 8 distinct cell clusters. **d** t-SNE map showing all identified clusters. Different colors indicate clusters, different shapes indicate stimulation time. **e** The employed workflow allowed to link protein expression data acquired during FACS to the transcriptome data. The number of IFNα+ cells assigned to each cluster, and the percentage of sorted IFNα+ cells in each cluster, is plotted against the cluster name. **f** t-SNE map showing the fluorescence intensity of IFNα and TNFα as measured during FACS for each cell. **g** Shown are transcript counts for genes of the type I IFN response and the TNF gene in single cells stimulated for 2 h with CpG-C. IFNα+ cells, identified during FACS, are indicated in blue, other cells are shown in red. **h** Genes that were upregulated in cluster 5 compared to cells from all other clusters were detected ($p <$ $10^{-8}$). Shown is the log2(fold change) for each gene. The color scale indicates the corresponding $p$ value

with neutralizing sera reduced the fraction of cells secreting IFNα to similar numbers as observed in our droplet experiments (Supplementary Fig. 14), indicative for a pool of early responder cells.

Based on our results, we propose the following model of early pDC activation (Fig. 6g). The pDCs are able to produce IFNα constitutively, but this is a rare and stochastic process that is controlled by transcription factors such as nuclear factor (NF)-κB

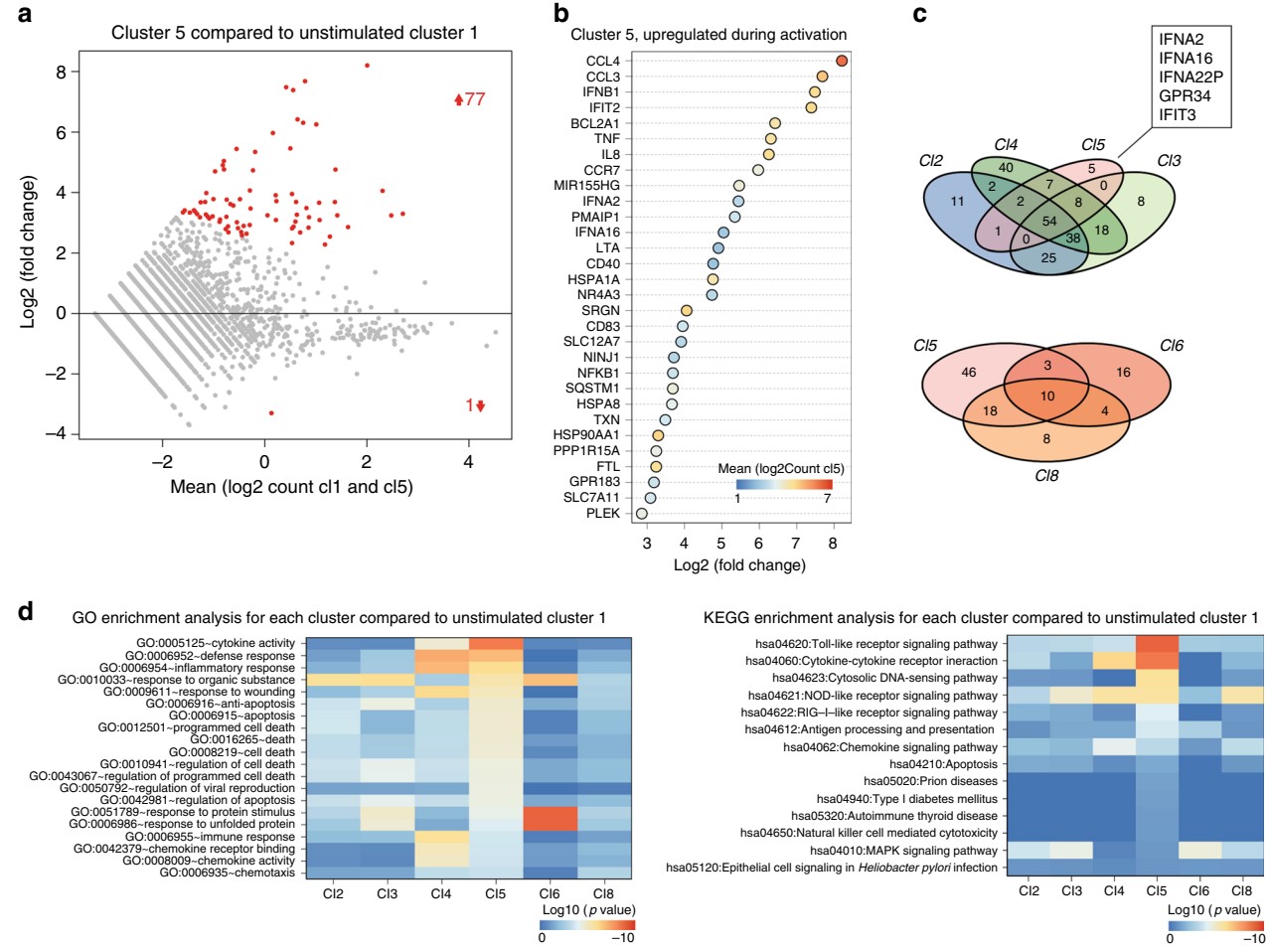

**Fig. 4** Type I IFN-expressing pDCs show unique gene expression patterns. **a** Differentially regulated genes in cells from all activated clusters compared to cluster 1 were identified. Differentially regulated genes from cluster 5. The average log2(count) of each gene is plotted against the log2(fold change) compared to cells in Cl1. Red color indicates $p$ value < $10^{-8}$. **b** The top 30 most upregulated genes are shown for cluster 5. Shown is the log2 fold change for each gene. The color scale indicates the average log2(Count) for each transcript in Cl5. **c** Venn diagram of the upregulated genes (log2(fold change) > 1.5, $p$ value < $10^{-8}$) within different clusters. **d** Lists of upregulated genes were submitted to DAVID for GO enrichment analysis and KEGG enrichment analysis. Heat maps show the most significantly enriched terms for the gene list from cluster 5. The color scale indicates the significance of enrichment of a particular term in all selected clusters after Benjamini–Hochberg correction for multiple testing

or activator protein 1 (AP-1), but not IRF7[27]. TLR triggering, which behaves as a sensitive and digital process, leads to the activation of the MyD88–IRF7 pathway and a 20-fold increase in stochastic IFNα expression. In many cells, this pathway is, however, limited by downstream components and by trafficking of CpG molecules to early endosomes. This leads to a still very limited pool of early responder pDCs that secrete type I IFN. Secreted type I IFN, then, primes surrounding pDCs and induces the expression of important factors for the IFNα production. This increases the probability of IFNα expression in those cells and leads to a robust population response.

## Discussion

We show that type I IFN production by freshly isolated human pDCs is controlled by stochastic gene regulation and amplified by environmental signals. This is supported by several observations. First, TLR signaling was necessary but insufficient for the induction of type I IFN production. Only a minor subset of cells produced IFNα, whereas all cells expressed TNFα or CCR7, implying universal activation of the TLR signaling pathway. Second, neither TLR signaling strength nor duration influenced

the fraction of IFNα-producing cells. Third, RNA profiling of single pDCs indicated no evidence of a privileged pDC subset with superior ability to produce type I IFN. On the contrary, a type I IFN-expressing pDC subset emerged at 2 h after activation, at the same time as type I IFN secretion was first observed in droplets, indicating that heterogeneity emerges simultaneously at protein and messenger RNA (mRNA) levels. Stochastic gene regulation is one of the strongest drivers of cellular heterogeneity and can be caused by not only the inherently random nature of gene expression itself but also by limitations in the signaling pathways leading to the production of a cytokine[18,19,21,32]. In our system, high IRF7 expression, which represented the most important cause of stochasticity in other systems, did not guarantee type I IFN production in all pDCs[18].

In contrast, we show here that the microenvironment has a decisive impact on type I IFN responses as IFNα production by pDCs depended directly on cell density. This effect could be mimicked by pre-treating pDCs with type I IFN leading to an increase in the fraction of IFNα-producing individually stimulated pDCs. This combination of stochastic gene regulation and environmental response amplification poses an efficient yet

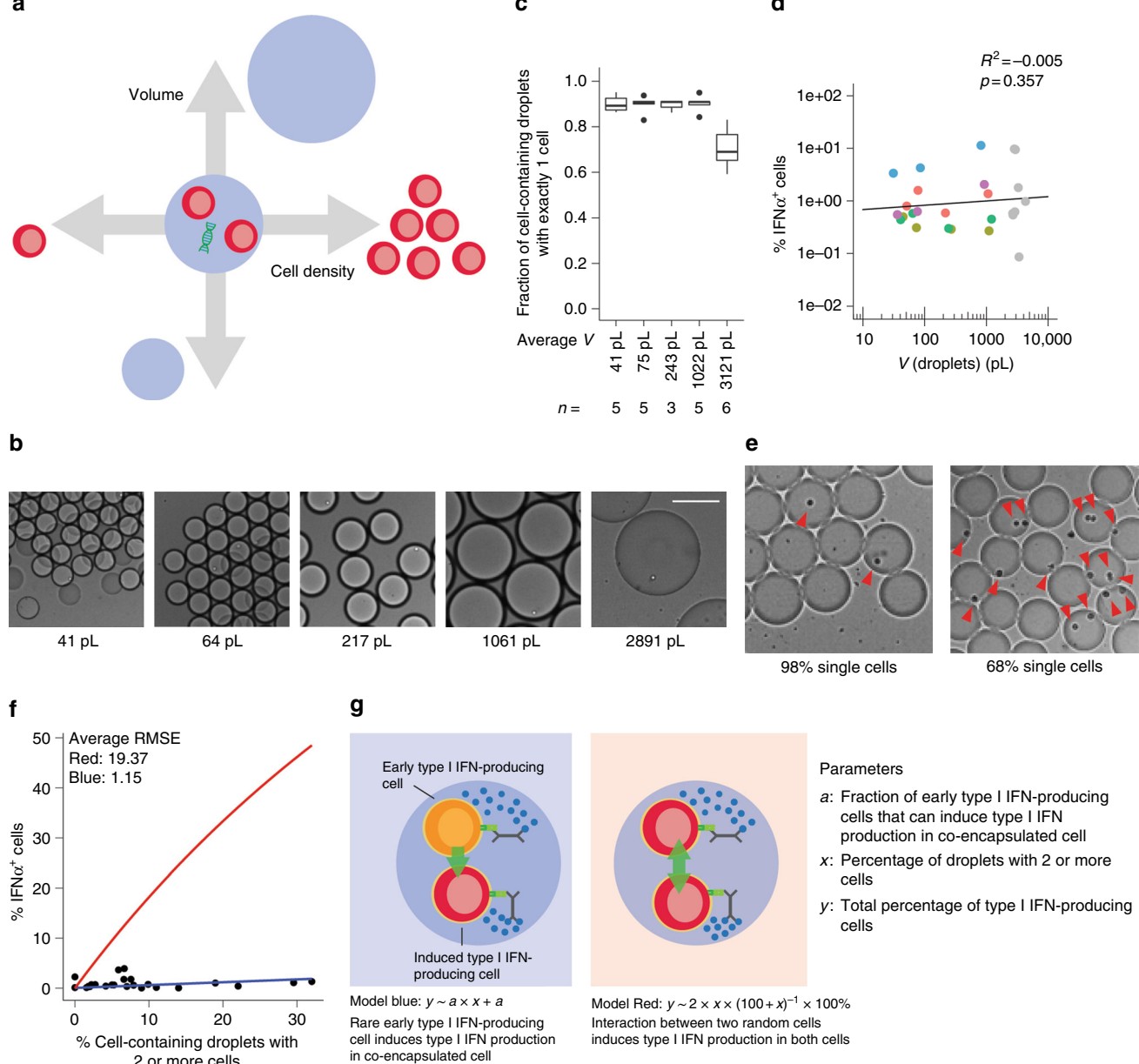

**Fig. 5** Microenvironmental factors influence the fraction of IFNα-producing pDCs. **a**, **b** The pDCs were coated with cytokine capture reagents, encapsulated in droplets of varying size (**b**, scale bar equals 100 μm), and stimulated individually with 50 μg/mL CpG-C for 12 h. The distribution of cells in droplets was measured by manual analysis of microscopic images after production. After staining for viability and surface marker expression, viable cells and TNFα-positive cells were detected via flow cytometry. **c** Shown is the fraction of cell-containing droplets with exactly one cell; $n \geq 3$. Lines indicate mean, hinges mark interquartile ranges, and whiskers reach to the highest/lowest value that is within 1.5 × interquartile range. **d** The fraction of IFNα-secreting pDCs was plotted against droplet volume. Dots of the same color indicate cells from the same donor. Gray dots are all originating from different donors. Linear regression was employed to calculate a trend line; $n = 24$. **a**, **e** The pDCs were stimulated in ~92 pL droplets with an increasing fraction of droplets containing more than 1 cell. **f** Shown is the fraction of IFNα-secreting cells plotted against the fraction of droplets containing more than 1 cell. Two models were generated to explain the observed pattern: (red) two random pDCs co-encapsulated in the same droplet can induce type I IFN production in each other; (blue) rare early type I IFN-producing cells can induce type I IFN production in conventional co-encapsulated cells. The root-square-mean error (RMSE) was calculated for both models to compare the fit to the data; $n = 24$. **g** Schematic overview of both models

flexible solution for pDCs to generate robust type I IFN responses. IFNα production by rogue cells that detect host-derived nucleic acids in sterile situations is limited without type I IFN amplification, but rapid and robust responses are guaranteed when pDCs are triggered in inflamed tissue with high pDC density or type I IFN signals from infected body cells. Furthermore, controlling type I IFN production in such a population-regulated stochastic manner allows the induction of an antiviral state in all cells of a given tissue but bypasses the necessity of all cells being type I IFN producers, reducing type I IFN levels and tissue damage.

These insights have far-reaching implications: on an applied level, pDC-focused treatments, such as DC-based immunotherapy, need to reconsider vaccine parameters, such as number of injected cells, location and pre-treatment of injection side, and cell density during stimulation for better efficacy. On a more fundamental level,

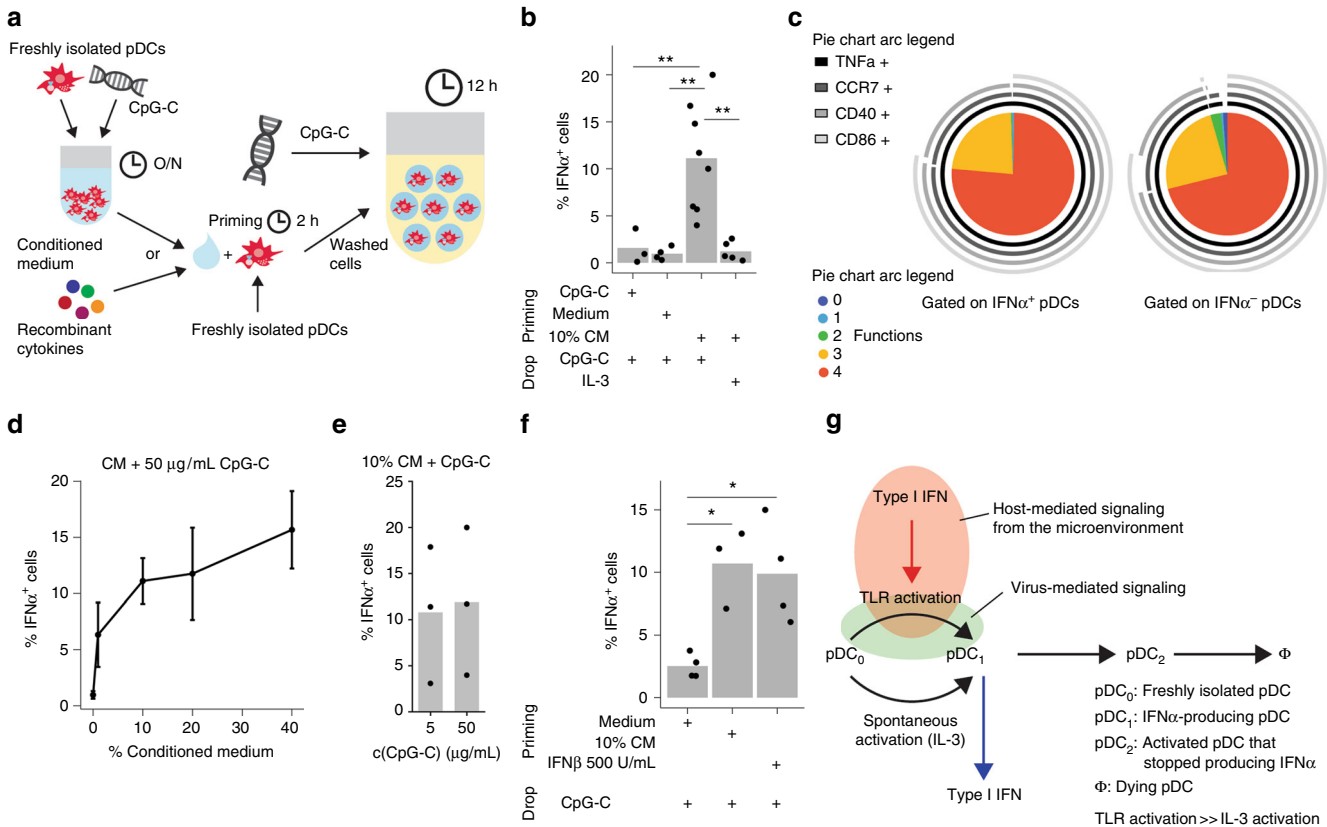

**Fig. 6** The pDCs primed with CM or IFNβ are more likely to produce IFNα. **a** Schematic overview of the priming experiment. The pDCs were stimulated for 2 h with conditioned medium (CM), 0.5 μg/mL CpG-C, or control medium. Cells were washed, coated with cytokine capture reagent, encapsulated in droplets, and stimulated individually with 0.01 μg/mL IL-3 or 50 μg/mL CpG-C for 12 h. CM was generated from bulk pDC cultures stimulated with 5 μg/mL CpG-C at a density of 25,000 cells/well. Cytokine-secreting cells were detected using flow cytometry. **b** The fraction of IFNα-secreting cells is plotted against different treatment conditions. **c** Co-expression of CCR7, CD40, CD86, and TNFα by single IFNα+ and IFNα− pDCs was analyzed. Shown is the relative contribution of each functional response pattern to the total pDC population. **d** The pDCs were primed with different concentrations of CM and stimulated with 50 μg/mL CpG-C. Shown is the fraction of IFNα-secreting cells plotted against CM concentration; $n \geq 3$. Dots indicate mean and whiskers indicate SEM. **e** The pDCs were treated as described above, primed with 10% CM, and stimulated with different concentrations of CpG-C. Shown is the fraction of IFNα-secreting cells plotted against CpG-C concentration; $n = 3$. **f** The pDCs were primed with 10% CM, control medium, or 500 U/mL IFNβ, and stimulated with 50 μg/mL CpG-C. The fraction of IFNα-secreting cells is plotted against different treatment conditions; $n \geq 3$. **g** Schematic model illustrating stochastic IFNα-production by pDCs. Few pDCs produce IFNα constitutively without stimulation by TLR ligands, here resembled by differentiation from a freshly isolated pDC (pDC$_0$) to an IFNα-secreting pDC (pDC$_1$). Literature indicates that this is not IRF7 dependent but NF-κB and AP-1[27]. Upon TLR9 triggering the IRF7-dependent pathway is activated which also allows differentiation to IFNα-secreting pDC at a much higher rate. Despite involvement of the IRF7 pathway, still only very few pDCs produce IFNα. Paracrine signaling via the type I IFN receptor can increase this rate, probably via the upregulation of IRF7 expression, leading to a large fraction of cells expressing IFNα. After producing IFNα, pDCs become refractory to re-stimulation (pDC$_2$) and eventually die (Φ). **b**, **f** Welch-corrected two-sample $t$-test; *$p < 0.05$, **$p < 0.01$

our insights imply that the functional behavior of pDCs is plastic and adaptable to local cues from the tissue microenvironment, similar to macrophages[42]. Therapy approaches that target pDCs inside the body should take into account that not all pDCs are the same and that pDCs might react differently to treatment depending on the tissue context of the disease.

Here, we show that well-studied human primary immune cell responses can be based on stochastic processes at the single-cell level and emphasize the importance of single-cell techniques to deconstruct immunological responses at the single-cell level.

## Methods

**Antibodies and cell stimuli**. For a full list of utilized antibodies and reagents, the readers are referred to the Supplementary Methods.

**Cell isolation and culture**. Jurkat T cells (ATCC, Clone E6-1 (ATCC® TIB-152™)) were cultured in RPMI (Thermo Fischer Scientific) supplemented with 10% fetal calf serum (FCS; Greiner Bio-One) and 1% Antibiotic-Antimycotic (Life Technologies), and regularly tested for mycoplasma contamination. The pDCs were

obtained from buffy coats of healthy donors (Sanquin) after written informed consent per the Declaration of Helsinki and according to institutional guidelines. In short, peripheral blood mononuclear cells (PBMCs) were isolated from donor blood via Ficoll density gradient centrifugation (Axis-Shield). The pDCs were subsequently isolated using magnet-activated cell sorting (MACS) or fluorescence-activated cell sorting (FACS).

For MACS isolation, PBMCs were resuspended in X-Vivo 15 cell culture medium (Lonza) supplemented with 2% pooled human serum (HS; Sanquin) and incubated for 1 h at 37 °C in cell culture flasks T75 (Corning) to deplete monocytes. Cells were washed thrice with phosphate-buffered saline (PBS; Braun) while non-adherent cells were collected. The pDCs were isolated from this cell population by positive selection using the CD304 Microbeat Kit (Miltenyi Biotec) following the manufacturer's instructions. Cells were counted and purity was assessed using flow cytometry. For this purpose, cells were washed with PBS supplemented with 0.5% bovine serum albumin fraction V (BSA, Roche) and 0.01% NaN$_3$ (Merck; subsequently referred to as PBA) and stained for 10 min at 4 °C using APC-labeled anti-CD303 and fluorescein isothiocyanate (FITC)-labeled anti-Lineage Cocktail 1 (LIN1) antibodies in 30 μL PBA. The pDCs were identified as CD303+LIN− and purity was on average 93% (Std.: 3.76%, $n = 67$).

For FACS isolation, PBMCs were washed and FITC-labeled anti-LIN1 antibodies were added to the pellet. Cells were incubated for 20 min at 4 °C. Subsequently, cells were washed with PBS supplemented with 4 mM

ethylenediaminetetraacetic acid (EDTA; Sigma) and 0.1% BSA (subsequently referred to as wash buffer) and anti-FITC microbeads (Miltenyi Biotec) were added to the pellet. Cells were incubated for 30 min at 4 °C and subsequently washed with wash buffer. LIN1-positive cells were magnetically depleted using an LD column (Miltenyi Biotec) following the manufacturer's instructions. Cells were washed with wash buffer and VioBlue- or PE-Cy7-labeled anti-HLA-DR and BV510- or PE-labeled anti-CD304 antibodies were added to the pellet. Cells were incubated for 30 min at 4 °C and afterwards washed with wash buffer. The pDCs were sorted as LIN1−HLA-DR+CD304+ cells on a FACS Aria II SORP (BD).

During stimulation, pDCs were cultured in X-Vivo 15 supplemented with 2% HS or RPMI supplemented with 10% FCS.

**Soft lithographic procedure**. The microfluidic device was molded against an SU-8 photo resist structure on a silicon wafer using a commercially available poly-dimethylsiloxane silicone elastomer (Sylgard 184, Dow Corning). The surface of the Sylgard 184 was OH-terminated by exposure to plasma (Diener Electronic GmbH), and was sealed with another plasma-treated glass cover slide to yield closed micro channels. Channels were treated with a 2% silane solution.

**Microfluidic setup**. Soft lithographic techniques were used to fabricate microfluidic channels (see above). Liquids were dispensed from plastic syringes (Becton Dickinson), which were connected to the microfluidic device by polytetrafluoroethylene tubing (Novodirect GmbH). The syringes were driven by computer-controlled syringe pumps (Nemesys, Cetoni GmbH). For the stability of droplets, 3 w/w% Pico-Surf® surfactant (Sphere Fluidics) was used in fluorinated HFE-7500 oil (Novec 7500, 3M). Cells and stimuli were loaded separately on the microfluidic chip. The dimensions of the microfluidic channels are 40 μm × 25 μm at the first inlet, 60 μm × 25 μm at the second inlet and the production nozzle, and 100 μm × 25 μm at the collection channel.

**Priming and blocking**. To block type I IFN signaling, pDCs were incubated at 37 °C for 30 min with medium containing blocking antibody against IFNAR2 (PBL Assay Science, 10 μg/mL) and neutralizing sera against IFNα and IFNβ (both from PBL Biomedical Laboratories, both 1000 NU/mL). To prime, pDCs were resuspended in medium containing cytokines or conditioned medium and incubated for 2 h, 37 °C. Subsequently, cells were washed thrice with wash buffer and prepared for downstream applications.

**Single-cell activation assay**. Cells were washed twice with wash buffer and incubated in 100 μL per 10$^6$ cells wash buffer containing Cytokine Catch Reagent (Miltenyi Biotec) at 4 °C for at least 40 min. Control experiments excluded that the employed Cytokine Catch Reagents affect viability or cellular functions (Supplementary Fig. 15). Next, cells were washed with wash buffer and medium and resuspended in medium at 2.6·10$^6$ cells/mL for single-cell encapsulation in 70–100 pL droplets. In case of experiments using different droplet sizes or multiple cells per drop, these concentrations were adjusted to yield the desired Poisson distribution. Stimulus was dissolved in medium at twice the desired concentration to account for on-Chip dilution. For 90 pL droplet production, flow rates were adjusted to 900 μL/h for the oil phase and 200 μL/h for the aqueous fractions (Supplementary Table 4) for overview of all employed flow rates). In all experiments, constant volumetric flow rates were used. To assess the encapsulation rate, videos of the droplet production and images of the produced emulsion were acquired using a CKX41 microscope (Olympus) at ×10 magnification. Encapsulation rate was manually assessed using Fiji[43,44]. The emulsion was collected and covered with medium to protect droplets from evaporation. Cells were incubated with open lid at 37 °C and 5% CO$_2$. Next, the emulsion was broken by adding 150 μL HFE-7500 containing 20% w/w 1H,1H,2H,2H-Perfluoro-1-octanol and centrifuging briefly at 60 relative centrifugal force (RCF). The cell-containing aqueous phase was transferred into a new tube containing 500 μL PBA and left for 2 min to allow residual oil to settle. Finally, the aqueous phase was transferred into a clean tube and cells were washed with PBS. Dead cells were identified by staining with Fixable Viability Dye eFluor® 780 (eBioscience, 1:2000 in PBS, 100 μL) for 30 min at 4 °C. Cells were washed once with PBS and blocked with PBA supplemented with 1% HS for 10 min at 4 °C. To stain for surface proteins and captured cytokines, cells were incubated with antibodies in 70 μL PBA supplemented with 1% HS for 10 min on ice. After incubation, cells were washed and resuspended in PBA and kept at 4 °C until acquisition on a FACS Verse flow cytometer (BD).

**RNA isolation and quantitative PCR**. RNA was isolated using Trizol (Life Technologies) following the manufacturer's protocol. RNA quantity was determined on NanoDrop 2000c (Thermo Scientific) and RNA quality was determined via agarose gel electrophoresis. Then, 2 μg of RNA was DNAse I-treated to remove residual genomic DNA and reverse transcribed into complementary DNA (cDNA) by M-MLV reverse transcriptase (Life Technologies) to obtain 25 μL of cDNA. The cDNA was diluted 25× in nuclease free water. For each reaction, 4 μL diluted cDNA, 300 nM primers, 10 μL SYBR Green (Roche), and water were added to a final volume of 20 μL. Each sample was amplified using a CFX96 sequence detection system (Bio-Rad). The following quantitative PCR (qPCR) cycling conditions were used: 50 °C/2 min, 95 °C/10 min, 40 cycles of 95 °C/15 s; 60 °C/1 min,

melt analysis 60 °C–95 °C with increment 0.5 °C/5 s. The gene-specific oligonucleotide primers used to determine the expression of the genes of interest are listed in Supplementary Table 2. To increase the chance of consistency, qPCR primers were based on the MA probes with highest differential expression. PCR products were monitored by measuring the increase of fluorescence caused by binding of SYBR Green. Quantitative PCR data were analyzed using CFX96 manager and relative expression of the gene of interest was determined using the cycle threshold method with GAPDH as reference genes[45].

**Perturbation profiling–scRNA-seq**. Using FACS as described above, single cells were sorted in 384-well plates containing a 50 nL droplet with CELseq2-primers and covered by mineral oil. A Mosquito® HTS (TTP Labtech) was used to dispense the droplets. To remove red blood cells, PBMCs were resuspended in 8 mL of ice-cold ACK buffer and incubated for 5 min on ice prior antibody staining with FITC-labeled anti-LIN1. Subsequently, the cells were washed with X-Vivo 15 supplemented with 2% HS and the standard protocol was further followed.

After sorting, plates were immediately frozen at −80 °C until further processing. Several days later, plates were thawed and incubated at 65 °C for 5 min to lyse cells. Perturbation profiling was conducted using the SORT-Seq protocol[33]. In short, spike-in RNA, reverse transcriptase and second-strand mixes were added to the wells using the Nanodrop II liquid handling platform (GC Biotech). Subsequently, the mRNA of each cell was reverse transcribed and converted to double-stranded cDNA. Libraries were then pooled, and in vitro transcribed for linear amplification, following the CEL-Seq 2 protocol[46]. Primers consisted of a 24 bp polyT stretch, a 6 bp random molecular barcode (unique molecular identifier (UMI)), a cell-specific 8 bp barcode, the 5′ Illumina TruSeq small RNA kit adaptor, and a T7 promoter. Illumina sequencing libraries were then prepared with the TruSeq small RNA primers (Illumina) and sequenced paired-end at 75 bp read length (high output) on the Illumina NextSeq.

**Stimulation in microtiter plate**. The pDCs were resuspended in 100 μL medium containing the appropriate stimulus (see supplementary methods) and cultured in 96-well round bottom plates (Costar, polystyrene) at a density of 25,000 cells per well if not stated differently. Depending on the experimental setting, Brefeldin A (Sigma, 10 μg/mL) was added 2 h before harvesting the cells.

**Antibody staining**. Cells were washed once with PBS and dead cells were identified by staining with Fixable Viability Dye eFluor® 780 (eBioscience, 1:2000 in PBS, 100 μL) at 4 °C for 30 min. Subsequently, cells were washed once with PBS and blocked with PBA supplemented with 1% HS at 4 °C for 10 min. Cells were washed and incubated with antibodies against surface proteins in 30 μL PBA supplemented with 1% HS for 10 min on ice. Afterwards, cells were washed with PBA followed by a wash with PBS. Cells were fixed and permeabilized with Cytofix/Cytoperm solution (BD, 100 μL) for 20 min at 4 °C. Next, cells were washed with Perm/Wash buffer (BD) and blocked for 10 min at 4 °C using Perm/Wash buffer supplemented with 1% HS. Subsequently, cells were incubated with antibodies against intracellular proteins in 30 μL Perm/Wash buffer supplemented with 1% HS for 30 min at 4 °C. Cells were washed twice with Perm/Wash buffer followed by a wash with PBA and resuspended in PBA. For IRF7 staining, cells were instead fixed with 4% paraformaldehyde (Merck) in PBS for 10 min at room temperature. After incubation, PBA was added and cells were washed twice with PBA, followed by a wash with PBA supplemented with 0.1% Triton X (Sigma). Cells were blocked for 10 min at 4 °C using PBA supplemented with 0.1% Triton X and 1% HS. Subsequently, cells were incubated with antibodies against intracellular proteins in 30 μL PBA supplemented with 0.1% Triton X and 1% HS for 30 min at 4 °C. Cells were washed twice with PBA supplemented with 0.1% Triton X followed by a wash with PBA and resuspended in PBA. All cells were kept at 4 °C until acquisition on a FACS Verse flow cytometer (BD). To guarantee highest purity in experiments, we limited our analysis to viable CD14−CD19− pDCs.

**ELISA analysis**. The enzyme-linked immunosorbent assay (ELISA) plates (Nunc MaxiSorp ELISA Plates for IFNα, Greiner bio-one high binding microplates for TNFα ELISA) were incubated with PBS containing anti-cytokine antibodies at the manufacturer-recommended concentration (Human IFN-alpha matched antibody pairs, Human TNF alpha ELISA Ready-SET-Go, both from eBioscience) overnight at 4 °C. Next, plates, coated with antibodies against IFNα, were washed once with PBS supplemented with 0.05% Tween-20 (Merck, subsequently referred to as ELISA wash buffer and used for all wash steps) and blocked using 250 μL ELISA wash buffer supplemented with 0.5% BSA for 2 h at room temperature. Plates were washed twice and incubated with 50 μL of sample or standard and 50 μL of horseradish peroxidase (HRP)-conjugated detection antibody at the recommended concentration for 2 h at room temperature. Plates coated with antibodies against TNFα were washed once and blocked with ELISA dilutent (eBioscience) for 2 h at room temperature. Plates were washed once and incubated with 50 μL sample or standard. Next, plates were washed 4× and incubated with detection antibody at the recommended concentration. Subsequently, plates were washed 4× and incubated with Avidin-HRP at the recommended concentration for 30 min at room temperature. Finally, all plates were washed trice and incubated with 100 μL TMB

Solution (eBioscience). Reaction was stopped by adding 100 μL of 1 M $H_3PO_4$ and absorption was measured at 450 nm using a microplate reader (Bio-Rad).

**Flow cytometry and ELISA analysis**. Flow cytometry data were analyzed using FlowJo X (Tree Star) and SPICE (downloaded from http://exon.niaid.nih.gov)[47]. Analysis and presentation of distributions was performed using PRISM for windows version 5.03 (GraphPad) and The R Project for Statistical Computing using the ggplot2, reshape, and xlsx packages[48–51]. For statistical analysis, Student's t-test, Mann–Whitney test, and linear regression analysis using least square fit were performed.

**Linear regression model**. Two models were generated: direct interactions between two random pDCs amplify the type I IFN production (2 × percentage of droplets with >1 cells/[100% + percentage of droplets with >1 cells] × 100% ~ percentage of cells that produce IFNα); interactions between early type I IFN-producing pDCs and other pDCs amplify type I IFN production (percentage of droplets >1 cells · percentage of early-responding cells + the percentage of early-responding cells ~ percentage of cells that produce IFNα). In both models, droplets with 3 or more cells are treated as if they contained only 2 cells. To compare the fit of each model, the dataset (n = 24) was randomly split into training (75%) and test (25%). Model parameters were estimated based on the training dataset, and the test dataset was used to predict the fraction of type I IFN-producing cells. Predicted and measured values were compared using the root-mean-square error (RMSE). This process was repeated 100 times and the average RMSE for each model was calculated.

**ScRNA-seq analysis**. Paired-end reads from Illumina sequencing were aligned to the human transcriptome with BWA[52]. Read 1 was used for assigning reads to correct cells and libraries, while read 2 was mapped to gene models. Reads that mapped equally well to multiple locations were discarded. Read counts were first corrected for UMI barcode by removing duplicate reads that had identical combinations of library, cellular, and molecular barcodes and were mapped to the same gene. Transcript counts were then adjusted to the expected number of molecules based on counts, 4096 possible UMIs, and poissonian counting statistics.

Samples were normalized by down-sampling to a minimum number of 1700 transcripts. RaceID2 was used to cluster cells and to perform outlier analysis. Differentially expressed genes between two subgroups of cells were identified based on DEseq[33]. Gene ontology (GO) and KEGG (Kyoto Encyclopedia of Genes and Genomes) analysis was conducted by submitting lists of up to 50 most upregulated genes (log2(fold change) of >1.5, adjusted p value < $10^{-8}$) to the DAVID 6.7 online platform[40,41].

**Data availability**. All relevant data related to this manuscript are available on request from the authors. The accession number for the single-cell RNA-sequencing data described in this study is GEO: GSE114161. All relevant codes related to this manuscript are available from the authors or as Supplementary Information.

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

## Acknowledgements

The authors want to thank Johannes Textor and Mauro Muraro for supporting the statistical analysis and Michael Valente for enthusiastic discussions. F.W. was supported by a Radboudumc PhD grant. J.T. is supported by NWO-Veni grant 863.130.24 from the Netherlands Organization for Scientific Research and acknowledges generous support by the Eindhoven University of Technology. I.J.M.d.V. received an NWO-Vici award 918.146.55. C.G.F. received the NWO Spinoza award, ERC Advanced Grant PATHFINDER (69019), and Dutch Cancer Society KWO grant 2009–4402. W.T.S.H. acknowledges generous support by the Radboud University and funding from the Ministry of Education, Culture and Science (Gravity program 024.001.035).

## Author contributions

F.W. designed and performed experiments, analyzed results, and wrote the manuscript. N.S., N.v.B., D.H. and J.V. performed experiments and analyzed results. I.B.-R., R.W., A.P. and J.F.M.J. performed experiments. A.v.O. and H.D. supervised the research. C.G. F., W.T.S.H. and I.J.M.d.V. supervised the research and wrote the manuscript. J.T. designed and performed experiments, analyzed results, supervised the research, and wrote the manuscript.

## Additional information

**Competing interests:** The authors declare no competing interests.

