## [Peer Review File · Nature Communications]

Reviewers' comments:

Reviewer #1 (Remarks to the Author):

The authors described droplet based single cell (pDC) approach for cytokine secretion with single-cell RNA-seq profiling . The stimulation and capture of secreted cytokine was conducted in droplets. After breaking the emulsion, the cells were sorted and specific subpopulation was profiled via RNA-seq.

The capture of the secreted cytokine is conducted by "capture-reagents for cytokine readout", which is bound to the cells surface. Thus the cells serve as capture platform for the secreted cytokine. This process could change the biological activity of cells and change the secretion levels or profiles of the cytokines. Thus the system is not a true biomimetic microenvironment.

Previous works describing DC or other immune cell activation and functional single cell analysis in droplets (in 2013), were not cited. In addition, others developed protocols for downstream analysis (RNA-seq) of droplet based sorted cells (Adam Abate and David Weitz). Thus the innovation of the droplet based approach and the assay to analyze subpopulations of single cells in this paper is minimal.

Reviewer #2 (Remarks to the Author):

Wimmers et al. have developed a droplet-based microfluidic system that allows simultaneous detection of surface marker expression, cytokine secretion and transcriptional profiles of single cells at a robust and high throughput manner. The authors utilize the system to investigate the functional heterogeneity within a human plasmacytoid dendritic cell (pDC) population and show that only a minor proportion of cells are able to secrete type I IFN upon stimulation, a key functional characteristic of pDCs at the population-level. The authors argue that this observation is due to stochastic gene expression in single cells early after stimulation, rather than the existence of a privileged subset within the population prior to stimulation, as no distinct subset was observed in the scRNAseq data. Furthermore, the authors demonstrate that altering cell density and priming of pDCs can influence the probability of IFN α secretion by single cells.

The study represents a great advance in single cell functional analysis, with insightful findings in the field of pDC biology, and of the programs underlying cytokine secretion in general. It also provides a novel framework as a single cell biological reactor for functional testing of small samples in the future, e.g. from patient material. The findings are interesting, timely and the manuscript is well written in a clear and logical way, and indeed was a pleasure to read.

Major criticism:

1. The authors state "Functional heterogeneity during the type I IFN response emerges, thus, on the transcriptome level early during pDC activation and in parallel to type I IFN production on the protein level arguing against the presence of a preexisting subset of privileged pDCs". I'm not sure I buy this logic as the authors cannot exclude this as yet. As they indicate, it is conceivable that such a program is transcriptionally hidden or noisy. Alternatively, there could be differences in e.g. the phosphorylation status of certain signalling proteins in a subset of cells with transcriptional distinction. I think a safer statement and conclusion is something along the lines of "no obvious transcriptionally distinct pDC subset existed that could predict IFN-responsiveness. This could either be because IFN production is genuinely a stochastic process, or that the nature of such a privileged cell state cannot be determined a priori by the present technology. This could include the possibilities that"

2. While the droplet approach is innovative, I'm not convinced yet that a low-density culture experiment couldn't have achieved similar results (as has been done in Supp Fig 9). Could the authors demonstrate or better explain the advantage in the early stages of the manuscript?

Minor criticisms

1. How stable is the binding of catch reagent to the cell and/or cytokines? Can the binding break during de-emulsification?
2. Why was the scRNAseq experiment only performed in cells after 0, 1 and 2 hrs but not later, as figure 1H shows that the highest proportion of IFN α + cells is found after 6 hrs? As a result, the current dataset contains very few IFN α + cells and can miss possible heterogeneity in that population.
3. I am confused by line 458 which says, "Transcript counts were then adjusted to the expected number of molecules based on counts, 256 possible UMI's and poissonian counting statistics.". According to previous text, the length of the UMI is 6, which gives a maximum of $4^6=4096$ possible UMIs, not 256. It is unclear where the "256 possible UMI" come from.
4. Regarding the analysis related to figure 4A. I think it is fine to use z-value and fold change for DE analysis. But considering the high dropout and overdispersed nature of scRNA-seq data, it might be better to use method that is designed for RNA-seq data, such as edgeR or DEseq, or methods that are tailored for scRNA-seq data.
5. The two models presented in figure 5F&G are not well explained and thus very confusing. The corresponding legend should explain what each abbreviation (x, y & a) stands for. If I understand correctly, "a" stands for the probability of active cells that can activate co-encapsulated cells. If that is true, make this point absolutely clear in legends, main text and methods.
6. Both abstract and the last paragraph of introduction are written in a mix of past and present tense. When presenting interpretation of the study, present tense should be used rather than past tense.
7. Figure 1F: the label is confusing. Which plot represent viability and which represent CD14/CD19 level?
8. Figure 1K: the two pie charts are exactly the same...
9. Figure 2: no non-stimulating controls.
10. Figure 3: typo in title, "minor" not "minute".
11. Figure 3C: please label "Euclidean distance".
12. Figure 3E: show % sorted IFN α + cells in clusters as well, as all clusters have very different total number of cells.
13. Figure 5C: alignment of n is off.
14. Figure 6A: the schematic is not very clear. As multiple priming conditions are used but only conditioned medium is shown. Please also indicate how long is the stimulation in this experiment.
15. 4% ≥ 2
16. Side scatter, not sideward
17. Fig 1H label panels clearly on top of each panel, not small font in bottom left.
18. CD40/CCR7 comes up in DE list between C11 vs C15 – can you overlay expression of these markers by FACS on tSNE plots?
19. My suggestion would be to have Fig 3G first to clearly highlight where the protein levels are in these cells, before going into the gene expression correlates of protein.

Reviewer #3 (Remarks to the Author):

The study by Wimmers et al., nicely demonstrates how pDCs behave in the single cell situation and use the microfluidics technique to deduce response propagation in a cell population. Both the technique and data are novel and of interest to the community. The data in general is nicely presented and the conclusions seem mostly well founded. I do have some points of concern/

remarks:

1) A direct comparison between bulk analysis of cells from the same donor is never made. This comparison is important as it is the life line to existing literature on pDC IFN α production that have been mostly if not exclusively performed on bulk cultures. It is also an important control to keep track of the overall effect of culturing pDCs in this new system as compared to more conventional culture methods. In the supplemental data bulk culture cytokine secretion data is presented but it would be nice to also see data from the same donor analyzed by microfluidics and bulk culture side by side. This especially to assess whether the size of the IFN α secreting population in the microfluidics systems approaches that in the bulk culture upon Type I IFN or conditioned medium exposure (and whether blocking IFN α reduces the number of IFN α secreting cells in the bulk culture to that in the single cell microfluidics system, see also below).

2) Figures 2D (and possibly C). Likely data from the same donors were compared between these two time-points. In this case a paired analysis would be in place (which may render the difference in 2D significant....)

3) For the single cell RNA sequencing data; on how many shared genes was the clustering of cells in figure 3 performed? The text states that 774 cells were included expressing "an average of 902 unique genes". However these cells will not all express the same genes. On which genes was the clustering performed and if not on all how were these selected? Please provide the selection and identity of the genes and also the expression matrix used for clustering. The full RNA sequencing data and analysis should be deposited in a public database and/ or added to the manuscript as supplementary information. (This is mandatory for Nature research journals)

4) For the single cells experiment it is not indicated from how many individuals cells were derived. I therefore assume this means the analysis was performed on cells from a single donor (which should by all means be stated). Even though I acknowledge that analyzing cells from different donors together or side by side is complex and/ or costly, it is risky to fare on result from one donor only. An obscure viral infection in this individual for example could severely impact these results. Therefore either additional RNA single cell RNA sequencing should be performed on cells from additional donors or alternative that the most important observations from this analysis should be verified on other donors by other means.

5) Figure 3C. It is hard to trace back the clusters in the heat map. Only for the largest clusters this is clear. Adding clustering dendrograms would help. Please add a (unit) description to the heat map scale bar (from 0 to 1).

6) On line 151-153 it is written: "We argued that transcriptional differences between privileged pDCs and conventional pDCs might be too subtle to generate different cell clusters before the onset of type I IFN expression but may be visible in shared gene expression patterns." It is not clear to me what the authors mean by "but may be visible in shared gene expression patterns". From the subsequent description of data I deduce they authors mean that that IFN α producing cells may share gene expression patterns after stimulation? Please clarify

7) In the last part of the manuscript the authors increase the number of IFN α secreting cells by pre-conditioning with either medium from overnight bulk cultures or with IFN β . I have two questions about this part. First, why was IFN α itself not included in the cytokine screen and second, do the IFN α /b blocking antibodies in addition to reducing IFN α levels also reduce the number of IFN α producing cells in the bulk cultures to that of the microfluidics? If the conclusions of the authors on the amplification of TLR induced IFN α production by paracrine Type I IFNs is true, it should.

Reviewer #1 (Remarks to the Author):

The authors described droplet based single cell (pDC) approach for cytokine secretion with single-cell RNA-seq profiling. The stimulation and capture of secreted cytokine was conducted in droplets. After breaking the emulsion, the cells were sorted and specific subpopulation was profiled via RNA-seq. The capture of the secreted cytokine is conducted by "capture-reagents for cytokine readout", which is bound to the cells surface. Thus the cells serve as capture platform for the secreted cytokine. This process could change the biological activity of cells and change the secretion levels or profiles of the cytokines. Thus the system is not a true biomimetic microenvironment.

We strongly disagree with the reviewer. Importantly, our droplet-based cytokine-capture approach enables sensitive cytokine detection and makes no use of transport inhibitors, which negatively impact cellular function and viability. This enabled us to measure cytokine secretion for extended time periods in an accumulative rather than snapshot fashion and facilitated the analysis of extremely early secretion events within the first 30 minutes of stimulation. Early activation events are problematic to investigate with transport inhibitor-based methods as they negatively impact cell signaling thereby distorting the measurement. In contrast to microtiter-based approaches, our microfluidic setup makes use of a computer-controlled syringe pumps. This allowed us to precisely control environmental factors and vary droplet volume and local cell density in a range that currently cannot be obtained with conventional cultures. We have added this explanation on page 4 of the manuscript. Furthermore, the reviewer is concerned that the biological activity of the cells might be affected by using the cytokine capture constructs. The cytokine capture constructs are generated by Miltenyi biotec and commercially available. All Miltenyi capture constructs bind the cell membrane through CD45. To ensure that the biological activity of our cells is not negatively affecting the secretion of cytokines, pDCs were either coated with capture reagent, specific for IL-2 (a cytokine not produced by pDCs to avoid tampering with auto- and paracrine feedback in this test), or left untreated and stimulated with 5µg/mL CpG-C. After incubation cells were fixed, permeabilized, stained for cytokine expression and analyzed by flow cytometry. As expected, we observed that triggering CD45 did not affect the biological activity of pDCs and, most importantly, has no negative effect on type I IFN production.

Previous works describing DC or other immune cell activation and functional single cell analysis in droplets (in 2013), were not cited. In addition, others developed protocols for downstream analysis (RNA-seq) of droplet based sorted cells (Adam Abate and David Weitz). Thus the innovation of the droplet based approach and the assay to analyze subpopulations of single cells in this paper is minimal.

We thank the reviewer for the thorough reading of our manuscript. The major concern of the reviewer is the novelty of the here presented work. We are aware of the excellent work of Abate and Weitz groups (and we have cited this work in references 28, 29, 31). Furthermore, our data describes for the first time the unprobed heterogeneity of single human plasmacytoid DCs, a rather rare immune cell subset, at the here described depth which yielded the remarkable finding that only a very small portion of the population is responsible for the onset of type I IFN production. We have now included a more balanced section on this in the introduction of our improved manuscript.

“Generating thousands of identical droplets at high-throughput, allows massively parallelized single cell experiments within these bioreactors. Recent technological breakthroughs in the field of droplet-based microfluidics increased the throughput of single cell DNA and RNA-sequencing experiments by orders of magnitude.^{28, 29} Previous attempts by our lab and others to leverage this power for the analysis of cytokine secretion were hampered in their translation into practice due to complex detection equipment or difficult handling conditions.^{30, 31}”

Taken together, we feel that our droplet-based approach not only has an immunological but also technological impact for studying cellular heterogeneity on a single cell level. Probing this heterogeneity will likely lead to better understanding of immune cell functions and how to exploit functional ‘superior’ cell populations. This vision is shared by the NIH, which has invested over \$90 million aimed at developing tools and methods scientists need to study individual cells in cancer and biology during the last years.

Reviewer #2 (Remarks to the Author):

Wimmers et al. have developed a droplet-based microfluidic system that allows simultaneous detection of surface marker expression, cytokine secretion and transcriptional profiles of single cells at a robust and high throughput manner. The authors utilize the system to investigate the functional heterogeneity within a human plasmacytoid dendritic cell (pDC) population and show that only a minor proportion of cells are able to secrete type I IFN upon stimulation, a key functional characteristic of pDCs at the population-level. The authors argue that this observation is due to stochastic gene expression in single cells early after stimulation, rather than the existence of a privileged subset within the population prior to stimulation, as no distinct subset was observed in the scRNAseq data. Furthermore, the authors demonstrate that altering cell density and priming of pDCs can influence the probability of IFN α secretion by single cells.

The study represents a great advance in single cell functional analysis, with insightful findings in the field of pDC biology, and of the programs underlying cytokine secretion in general. It also provides a novel framework as a single cell biological reactor for functional testing of small samples in the future, e.g. from patient material. The findings are interesting, timely and the manuscript is well written in a clear and logical way, and indeed was a pleasure to read.

We thank the reviewer for the appreciation of our work, helpful suggestions and thorough reading of our manuscript.

Major criticism:

1. The authors state “Functional heterogeneity during the type I IFN response emerges, thus, on the transcriptome level early during pDC activation and in parallel to type I IFN production on the protein level arguing against the presence of a preexisting subset of privileged pDCs”. I’m not sure I buy this logic as the authors cannot exclude this as yet. As they indicate, it is conceivable that such a program is transcriptionally hidden or noisy. Alternatively, there could be differences in e.g. the phosphorylation status of certain signaling proteins in a subset of cells with transcriptional distinction. I think a safer statement and conclusion is something along the lines of “no obvious transcriptionally distinct pDC subset existed that could predict IFN-responsiveness. This could either be because IFN production is genuinely a stochastic process, or that the nature of such a privileged cell state cannot be determined a priori by the present technology. This could include the possibilities that ...”

We agree with the reviewer that this is indeed valid point. We believe that stringent subset thinking is reaching its boundaries and rapid adaptation to environmental stimuli, as we show in our manuscript, is the future. We also feel that an integrated multi-omic approach is the future for scientists that endeavor on probing cellular heterogeneity. Having said that, we do acknowledge the concern raised by the reviewer that at the current time it might be technically not possible to detect our described privileged subset a priori. Possibly, these boundaries include phosphorylation status or simply very subtle differences in expression of signaling molecules or even post-transcriptional modifications might be missed or overlooked by the current approach. We therefore have amended our manuscript accordingly and as suggested by the reviewer also added a statement on page 6 of our improved manuscript.

2. While the droplet approach is innovative, I’m not convinced yet that a low-density culture experiment couldn’t have achieved similar results (as has been done in Supp Fig 9). Could the authors demonstrate or better explain the advantage in the early stages of the manuscript?

We thank the reviewer for this remark and we agree that this deserves a clearer explanation in the manuscript to allow readers to comprehend the major advantages of our described approach. We have added a more balanced explanation on page 4 of our manuscript.

“Each droplet served as a standardized and independent cell reactor and allowed the investigation of tens of thousands of individually stimulated cells simultaneously. This massively parallel approach facilitated the characterization of rare, truly single cell behavior. The system greatly exceeds the throughput and possibilities

when compared to conventional limited dilution experiments which require numerous replicate cultures and, crucially, cannot prohibit cellular crosstalk. Let alone the enormous amount of culture plates, medium and pipetting. Further, the low droplet volume greatly reduced reagent consumption and allowed us to work with small numbers of (primary) cells. We routinely probed rare pDCs using as few as 40,000 cells as input, showing that our technique is highly suited for the use of small biological samples. Importantly, our droplet-based cytokine-capture approach enables sensitive cytokine detection and makes no use of transport inhibitors, which negatively impact cellular function and viability. This enabled us to measure cytokine secretion for extended time periods in an accumulative rather than snapshot fashion and facilitated the analysis of extremely early secretion events within the first 30 minutes of stimulation. Early activation events are problematic to investigate with transport inhibitor-based methods as they negatively impact cell signaling thereby distorting the measurement. In contrast to microtiter-based approaches, our microfluidic setup makes use of a computer-controlled syringe pumps. This allowed us to precisely control environmental factors and vary droplet volume and local cell density in a range that currently cannot be obtained with conventional cultures.”

Minor criticisms

1. How stable is the binding of catch reagent to the cell and/or cytokines? Can the binding break during de-emulsification?

We agree with the reviewer that it is essential to ensure the stability of the catch reagent during the de-emulsification process. We performed experiments where cells were labeled with the capture constructs and subsequently encapsulated in droplets together with an excess of soluble recombinant cytokines. Thereafter, during the de-emulsification process the cells were either retrieved using

- 1. the PFO-based approach described in the manuscript*
- 2. the PFO-based approach combined with the simultaneous addition of celltracker dye labelled cells that were not coated with the capture construct.*

Next, the samples were labeled with the detection antibodies and analyzed by flow cytometry. As we expected, none of the cell tracker dye labelled cells displayed any positive signal for the cytokine. These data directly imply that the catch reagents are stably bound on the cell membrane and are not affected by the de-emulsification process. Furthermore, the fact that we regularly observe close to 100% TNF α + cells after 12h in-drop incubation using our droplet assay indicates that the capture reagent binds tightly to the cells during breaking of emulsion. We are therefore convinced that our technical workflow has no implications on measuring cell function.

2. Why was the scRNAseq experiment only performed in cells after 0, 1 and 2 hrs but not later, as figure 1H shows that the highest proportion of IFN α + cells is found after 6 hrs? As a result, the current dataset contains very few IFN α + cells and can miss possible heterogeneity in that population.

We have observed that plasmacytoid DC ignite their type I IFN secretion machinery very early upon activation. The secretion of IFN α is as such readily measurable after 1-2 hours. Indeed, the reviewer is correct in suggesting that 6h single cell activation followed by scRNA-seq might yield a higher number of cells that are positive for IFN α , and for future research it might be interesting to investigate the potential heterogeneity in the 0.5-1% pDC population secreting IFN α . However, in our described approach we were especially interested in the early events during the onset of activation to try and identify the origin of the observed IFN α production heterogeneity. Therefore, to specifically address that research question, we intentionally only focused on 0, 1 and 2 hour time points.

3. I am confused by line 458 which says, “Transcript counts were then adjusted to the expected number of molecules based on counts, 256 possible UMI’s and poissonian counting statistics.”. According to previous text, the length of the UMI is 6, which gives a maximum of $4^6=4096$ possible UMIs, not 256. It is unclear where the “256 possible UMI” come from.

We thank the reviewer for pointing this out. Indeed, the maximum number of possible UMIs is 4096 and not 256. We have corrected this typo on line 479 accordingly.

4. Regarding the analysis related to figure 4A. I think it is fine to use z-value and fold change for DE analysis. But considering the high dropout and overdispersed nature of scRNA-seq data, it might be better to use method that is designed for RNA-seq data, such as edgeR or DEseq, or methods that are tailored for scRNA-seq data.

Following this suggestion from the reviewer we have used a script based on DEseq and re-analyzed our dataset accordingly. We have updated Figure 3H and Figure 4 after the newly performed analysis. We did not observe any major biological changes after re-analysis, but the statistics are indeed better now and the plots show minor changes.

5. The two models presented in figure 5F&G are not well explained and thus very confusing. The corresponding legend should explain what each abbreviation (x, y & a) stands for. If I understand correctly, “a” stands for the probability of active cells that can activate co-encapsulated cells. If that is true, make this point absolutely clear in legends, main text and methods.

We appreciate the comment that the explanation of the 2 models was rather brief. In order to substantially increase the clarity of the 2 models we have made a number of changes. First, we have simplified the model and we added extra information and explanation in an improved version of figure 5, together with an improved figure legend (page 27 of manuscript). Furthermore, we have made improvements to the red model to enable a more concise fit to the actual situation. Finally, in the M&M section on page 16/17 we have changed and improved the description of the linear regression model.

6. Both abstract and the last paragraph of introduction are written in a mix of past and present tense. When presenting interpretation of the study, present tense should be used rather than past tense.

We thank the reviewer for pointing this out. We have adjusted the corresponding sections accordingly.

7. Figure 1F: the label is confusing. Which plot represent viability and which represent CD14/CD19 level?

We have changed the label of the axis of Figure 1F to more clearly indicate that all markers are measured at the same time in the same fluorescence channel. We were faced with a limited number of fluorescence channels on the employed flow cytometer. To use these channels as efficiently as possible, we intentionally combined all undesirable events in a single dump channel: APC-H7. To achieve this, we first stained dead cells with the fixable viability dye eFluor780, which has an emission spectrum similar to APC-H7. After that, we stained contaminating B cells and monocytes with APC-H7-conjugated anti-CD19 and anti-CD14 antibodies, respectively. During analysis, dead cells, B cells and monocytes, all appeared positive in the same channel and were, for us, hence, indistinguishable. However, as we wanted to exclude all these events at once, we did not need to distinguish them from each other but only from viable pDC, which were APC-H7 negative.

8. Figure 1K: the two pie charts are exactly the same...

We thank the reviewer for the sharp observation. In the improved figure we now included the correct pie chart.

9. Figure 2: no non-stimulating controls.

Human plasmacytoid DCs die in culture when they are left untreated. Therefore, we cultured cells in the presence of IL-3, which in the field is considered as survival factor for plasmacytoid DCs, as non-stimulated control.

10. Figure 3: typo in title, “minor” not “minute”.

We have changed this in the improved manuscript.

11. Figure 3C: please label “Euclidean distance”.

We have changed this in the improved manuscript.

12. Figure 3E: show % sorted IFN α cells in clusters as well, as all clusters have very different total number of cells.

We have changed this and added a corresponding plot in Figure 3E in the improved manuscript.

13. Figure 5C: alignment of n is off.

We have changed this in the improved manuscript.

14. Figure 6A: the schematic is not very clear. As multiple priming conditions are used but only conditioned medium is shown. Please also indicate how long is the stimulation in this experiment.

To increase the clarity of Figure 6A we have added all employed conditions to the cartoon and indicated the incubation times clearly at all steps.

15. 4% ≥ 2

In our manuscript we refer at 1 location to ‘4%’, and that is in the M&M section where we describe the intracellular cytokine staining and usage of 4% paraformaldehyde for fixation of cells. In that respect we are unsure what the reviewer is referring to.

16. Side scatter, not sideward

We have changed this in the improved manuscript.

17. Fig 1H label panels clearly on top of each panel, not small font in bottom left.

We have changed this in the improved manuscript.

18. CD40/CCR7 comes up in DE list between C11 vs C15 – can you overlay expression of these markers by FACS on tSNE plots?

Unfortunately, the suggested tSNE plots cannot be generated as in the sorting experiments for scRNA-seq we did not include the measurement of these markers.

19. My suggestion would be to have Fig 3G first to clearly highlight where the protein levels are in these cells, before going into the gene expression correlates of protein.

This is an interesting suggestion and after changing this accordingly we agree that this is clearer.

Reviewer #3 (Remarks to the Author):

The study by Wimmers et al., nicely demonstrates how pDCs behave in the single cell situation and use the microfluidics technique to deduce response propagation in a cell population. Both the technique and data are novel and of interest to the community. The data in general is nicely presented and the conclusions seem mostly well founded. I do have some points of concern/ remarks:

We thank the reviewer for the kind words, helpful suggestions and thorough reading of our manuscript.

1) A direct comparison between bulk analysis of cells from the same donor is never made. This comparison is important as it is the life line to existing literature on pDC IFN α production that have been mostly if not exclusively performed on bulk cultures. It is also an important control to keep track of the overall effect of culturing pDCs in this new system as compared to more conventional culture methods. In the supplemental data bulk culture cytokine secretion data is presented but it would be nice to also see data from the same donor analyzed by microfluidics and bulk culture side by side. This especially, to assess whether the size of the IFN α secreting population in the microfluidics systems approaches that in the bulk culture upon Type I IFN or conditioned medium exposure (and whether blocking IFN α reduces the number of IFN α secreting cells in the bulk culture to that in the single cell microfluidics system, see also below).

This is a very good point and indeed, to the best of our knowledge, all studies with human pDCs have studied IFN α secretion in bulk. Although we previously performed numerous single cell experiments (total n>50) alongside bulk experiments (total n=14), we do understand that addressing this comment is important for the community to have a clear 1-to-1 comparison and to appreciate the results described in the manuscript. The suggested experiments are technically challenging and very lengthy (to warrant cell viability and function, as these start to decrease from the moment a buffy coat is drawn). Nevertheless, we have performed an additional set of independent experiments (n=6) with different healthy donors to assess the capacity of single pDCs to secrete IFN α / TNF α and compared it to the secretion of IFN α / TNF α of bulk stimulated pDCs from the same donor. The results of the new experiments are in accordance with the data presented in our manuscript and demonstrate that the capacity of cells to secrete IFN α on the single cell level is minimal compared to the cells that were stimulated in bulk (figures below; left IFN α , right TNF α). We have added these new data as an additional supplementary figure to the manuscript.

For the part on blocking type I IFN in bulk see comment 7 below.

2) Figures 2D (and possibly C). Likely data from the same donors were compared between these two time-points. In this case a paired analysis would be in place (which may render the difference in 2D significant....)

We appreciate the reviewers' suggestion, however the data that are presented in Figure 2D cannot be subjected to paired analysis. In these experiments we started with two independent experiments and incubated one for 12h and the other for 24h. Therefore, we don't sample the same population twice at different time points but we have independent populations of cells that were just derived from the same donor. As such a paired analysis cannot be performed.

3) For the single cell RNA sequencing data; on how many shared genes was the clustering of cells in figure 3 performed? The text states that 774 cells were included expressing “an average of 902 unique genes”. However these cells will not all express the same genes. On which genes was the clustering performed and if not on all how were these selected? Please provide the selection and identity of the genes and also the expression matrix used for clustering. The full RNA sequencing data and analysis should be deposited in a public database and/or added to the manuscript as supplementary information. (This is mandatory for Nature research journals)

We adapted the text in the manuscript for improved clarity. The clustering was performed on all cells and all genes present in the dataset after standard QC filtering (described in M&M). We thus did NOT make a gene selection but we used 774 cells expressing 13,214 genes for clustering. We have added a more balanced description of the clustering and changed the text in our results section accordingly.

All RNA sequencing data was submitted to be deposited in the GEO database. We currently await the accession number and will mention this number in the main manuscript. The R scripts used for the data analysis as well as tables with differentially expressed genes are attached as supplementary information.

4) For the single cells experiment it is not indicated from how many individuals cells were derived. I therefore assume this means the analysis was performed on cells from a single donor (which should by all means be stated). Even though I acknowledge that analyzing cells from different donors together or side by side is complex and/ or costly, it is risky to fare on result from one donor only. An obscure viral infection in this individual for example could severely impact these results. Therefore, either additional RNA single cell RNA sequencing should be performed on cells from additional donors or alternative that the most important observations from this analysis should be verified on other donors by other means.

We agree with the reviewer that the analysis of multiple donors for scRNA-seq is desirable. Indeed, obscure viral infections could have implications for the results although our cells were derived from a perfectly healthy donor (now also mentioned in the manuscript). Despite the great complexity and high costs of scRNA-seq experiments (as pointed out correctly by the reviewer), we have performed 2 additional experiments with cells from 2 additional independent donors at steady state. The results are displayed in Supp Figure 8 of our improved manuscript. We observed that all 3 donors clustered together, indicative for the robustness of our approach.

5) Figure 3C. It is hard to trace back the clusters in the heat map. Only for the largest clusters this is clear. Adding clustering dendrograms would help. Please add a (unit) description to the heat map scale bar (from 0 to 1).

We appreciate the reviewers comment that it is difficult to trace the clusters in the heat map. To increase the clarity of Figure 3C we improved the heat map to augment the visibility of all the clusters and added this to our manuscript.

6) On line 151-153 it is written: “We argued that transcriptional differences between privileged pDCs and conventional pDCs might be too subtle to generate different cell clusters before the onset of type I IFN expression but may be visible in shared gene expression patterns.” It is not clear to me what the authors mean by “but may be visible in shared gene expression patterns”. From the subsequent description of data I deduce they authors mean that that IFN α producing cells may share gene expression patterns after stimulation? Please clarify

The reviewer is correct, we indeed wanted to describe that the uniqueness of the pDCs in cluster 5 might become evident and that they share gene expression patterns amongst each other by comparing cells in cluster 5 to the differential gene expression profiles of all stimulated pDC clusters. To increase the clarity of this intended message we have clarified the corresponding section in the manuscript.

7) In the last part of the manuscript the authors increase the number of IFN α secreting cells by pre-conditioning with either medium from overnight bulk cultures or with IFN β . I have two questions about this part. First, why

was IFN α itself not included in the cytokine screen and second, do the IFN α /b blocking antibodies in addition to reducing IFN α levels also reduce the number of IFN α producing cells in the bulk cultures to that of the microfluidics? If the conclusions of the authors on the amplification of TLR induced IFN α production by paracrine Type I IFNs is true, it should.

We agree with the reviewer that IFN α would be a logical candidate to include in our screen, however, to exclude the possibility that any increase in signal might be caused by residual trace amounts of IFN α left after priming we opted not for including IFN α in the cytokine screen.

Concerning the second part of the comment, we agree with the reviewer that it would be interesting to perform blocking experiments in bulk cultures. Therefore, we have performed an additional set of independent experiments (n=5) with different healthy donors with 2 distinct approaches. In the first set of experiments pDCs were preincubated with IFNAR blocking antibodies and plated in medium containing neutralizing sera against IFN α and IFN β prior to activation. Subsequently, cells were activated in bulk as previously described and IFN α secretion was inferred from intracellular cytokine staining after 6 or 8 hours of activation. Secondly, as the intracellular cytokine staining provides only a snapshot in time we also opted for labeling cells with the capture construct (similar to the droplet approach) and thereafter followed similar steps as in the first set of experiments, so blocking IFNAR and neutralizing sera prior to activation. In these second set of experiments we activated cells in bulk and captured secreted cytokines for 14 hours in the presence or absence of blocking. Our experiments revealed that blocking the IFNAR in combination with neutralizing sera for IFN α and IFN β reduced the fraction of cells secreting IFN α to similar numbers as we observed in our droplet experiments (Figures: blocking type I IFN followed by [left] intracellular cytokine staining, [right] capture construct-based detection). Thus, these additional sets of experiments contribute to our earlier findings and underscore the importance of the amplification of TLR-induced IFN α response by paracrine type I IFN signaling. We have added these additional experiments as a new supplementary figure to our manuscript.

Reviewers' comments:

Reviewer #1 (Remarks to the Author):

The authors described secretion based system which cannot capture the true in vivo function of cells and in fact changes the cellular behavior thus present artificial system. My concern regarding "capture-reagents for cytokine readout", which is bound to the cells surface was not undressed by the answer : "...makes no use of transport inhibitors, which negatively impact cellular function...". This needs to be determined experimentally by conducting control experiments. Furthermore due to the other droplet based approaches for secretion and RNA detections , I think that the novelty of this droplet based concept/approach is not present.

Reviewer #2 (Remarks to the Author):

I am satisfied with the concerted efforts of the authors to address all of my concerns.

Reviewer #3 (Remarks to the Author):

The authors have answered to all my concerns and I have no further comments

Point-to-point reply to the Reviewer comment:

Reviewer #1 (Remarks to the Author):

The authors described secretion based system which cannot capture the true in vivo function of cells and in fact changes the cellular behavior thus present artificial system. My concern regarding "capture-reagents for cytokine readout", which is bound to the cells surface was not addressed by the answer : "...makes no use of transport inhibitors, which negatively impact cellular function...". This needs to be determined experimentally by conducting control experiments.

We understand the remaining concern that was raised by the reviewer and we agree that the addition of results describing these experiments will increase the clarity of our manuscript. Therefore, we have now added these data as a new supplementary figure 15, which we also added here, and described them accordingly in the new and improved version of our manuscript.

Supp Figure 15 – Effect of Cytokine Catch Reagents on cellular function and viability of bulk cultured pDCs. PDCs were coated with capture reagent or left untreated and subsequently activated with 5 μ g/mL CpG-C in microtiter plates for 6h or 8h at a density of 25.000 cells/well. A) IFN α - and TNF α -secreting cells were detected via intracellular cytokine staining and flow cytometry after 6 hours and the result of 1 representative donor is shown. B,C) Shown is the fraction of IFN α - or TNF α -secreting cells plotted against treatment condition and stimulation or either 6 hours or 8 hours. Circles indicate mean, error bars indicate SEM, n=5. D) The expression of CCR7, CD40 and CD86 on differently treated pDCs after 8 hours of activation, one representative experiment is shown. E) Shown is the viability and the expression of CCR7, CD40 and CD86 positive cells plotted against treatment condition and stimulation or either 6 hours or 8 hours. Circles indicate mean, error bars indicate SEM, n=3.